# A Recurrent Latent Variable Model
# for Sequential Data

**Junyoung Chung, Kyle Kastner, Laurent Dinh, Kratarth Goel,**
**Aaron Courville, Yoshua Bengio**[*]
Department of Computer Science and Operations Research
Université de Montréal
[*]CIFAR Senior Fellow
{firstname.lastname}@umontreal.ca

## Abstract

In this paper, we explore the inclusion of latent random variables into the hidden state of a recurrent neural network (RNN) by combining the elements of the variational autoencoder. We argue that through the use of high-level latent random variables, the *variational RNN* (VRNN)[1] can model the kind of variability observed in highly structured sequential data such as natural speech. We empirically evaluate the proposed model against other related sequential models on four speech datasets and one handwriting dataset. Our results show the important roles that latent random variables can play in the RNN dynamics.

## 1 Introduction

Learning generative models of sequences is a long-standing machine learning challenge and historically the domain of dynamic Bayesian networks (DBNs) such as hidden Markov models (HMMs) and Kalman filters. The dominance of DBN-based approaches has been recently overturned by a resurgence of interest in recurrent neural network (RNN) based approaches. An RNN is a special type of neural network that is able to handle both variable-length input and output. By training an RNN to predict the next output in a sequence, given all previous outputs, it can be used to model joint probability distribution over sequences.

Both RNNs and DBNs consist of two parts: (1) a transition function that determines the evolution of the internal hidden state, and (2) a mapping from the state to the output. There are, however, a few important differences between RNNs and DBNs.

DBNs have typically been limited either to relatively simple state transition structures (e.g., linear models in the case of the Kalman filter) or to relatively simple internal state structure (e.g., the HMM state space consists of a single set of mutually exclusive states). RNNs, on the other hand, typically possess both a richly distributed internal state representation and flexible non-linear transition functions. These differences give RNNs extra expressive power in comparison to DBNs. This expressive power and the ability to train via error backpropagation are the key reasons why RNNs have gained popularity as generative models for highly structured sequential data.

In this paper, we focus on another important difference between DBNs and RNNs. While the hidden state in DBNs is expressed in terms of random variables, the internal transition structure of the standard RNN is entirely deterministic. The only source of randomness or variability in the RNN is found in the conditional output probability model. We suggest that this can be an inappropriate way to model the kind of variability observed in highly structured data, such as natural speech, which is characterized by strong and complex dependencies among the output variables at different

timesteps. We argue, as have others [4, 2], that these complex dependencies cannot be modelled efficiently by the output probability models used in standard RNNs, which include either a simple unimodal distribution or a mixture of unimodal distributions.

We propose the use of high-level latent random variables to model the variability observed in the data. In the context of standard neural network models for non-sequential data, the variational autoencoder (VAE) [11, 17] offers an interesting combination of highly flexible non-linear mapping between the latent random state and the observed output and effective approximate inference. In this paper, we propose to extend the VAE into a recurrent framework for modelling high-dimensional sequences. The VAE can model complex multimodal distributions, which will help when the underlying true data distribution consists of multimodal conditional distributions. We call this model a *variational RNN* (VRNN).

A natural question to ask is: how do we encode observed variability via latent random variables? The answer to this question depends on the nature of the data itself. In this work, we are mainly interested in highly structured data that often arises in AI applications. By highly structured, we mean that the data is characterized by two properties. Firstly, there is a relatively high signal-to-noise ratio, meaning that the vast majority of the variability observed in the data is due to the signal itself and cannot reasonably be considered as noise. Secondly, there exists a complex relationship between the underlying factors of variation and the observed data. For example, in speech, the vocal qualities of the speaker have a strong but complicated influence on the audio waveform, affecting the waveform in a consistent manner across frames.

With these considerations in mind, we suggest that our model variability should induce *temporal dependencies across timesteps*. Thus, like DBN models such as HMMs and Kalman filters, we model the dependencies between the latent random variables across timesteps. While we are not the first to propose integrating random variables into the RNN hidden state [4, 2, 6, 8], we believe we are the first to integrate the dependencies between the latent random variables at neighboring timesteps.

We evaluate the proposed VRNN model against other RNN-based models – including a VRNN model without introducing temporal dependencies between the latent random variables – on two challenging sequential data types: natural speech and handwriting. We demonstrate that for the speech modelling tasks, the VRNN-based models significantly outperform the RNN-based models and the VRNN model that does not integrate temporal dependencies between latent random variables.

## 2 Background

### 2.1 Sequence modelling with Recurrent Neural Networks

An RNN can take as input a variable-length sequence $\mathbf{x} = (\mathbf{x}_1, \mathbf{x}_2, \ldots, \mathbf{x}_T)$ by recursively processing each symbol while maintaining its internal hidden state $\mathbf{h}$. At each timestep $t$, the RNN reads the symbol $\mathbf{x}_t \in \mathbb{R}^d$ and updates its hidden state $\mathbf{h}_t \in \mathbb{R}^p$ by:

$$\mathbf{h}_t = f_\theta\left(\mathbf{x}_t, \mathbf{h}_{t-1}\right),\tag{1}$$

where $f$ is a deterministic non-linear transition function, and $\theta$ is the parameter set of $f$. The transition function $f$ can be implemented with gated activation functions such as long short-term memory [LSTM, 9] or gated recurrent unit [GRU, 5]. RNNs model sequences by parameterizing a factorization of the joint sequence probability distribution as a product of conditional probabilities such that:

$$p(\mathbf{x}_1, \mathbf{x}_2, \ldots, \mathbf{x}_T) = \prod_{t=1}^{T} p(\mathbf{x}_t \mid \mathbf{x}_{<t}),$$
$$p(\mathbf{x}_t \mid \mathbf{x}_{<t}) = g_\tau(\mathbf{h}_{t-1}),\tag{2}$$

where $g$ is a function that maps the RNN hidden state $\mathbf{h}_{t-1}$ to a probability distribution over possible outputs, and $\tau$ is the parameter set of $g$.

One of the main factors that determines the representational power of an RNN is the output function $g$ in Eq. (2). With a deterministic transition function $f$, the choice of $g$ effectively defines the family of joint probability distributions $p(\mathbf{x}_1, \ldots, \mathbf{x}_T)$ that can be expressed by the RNN.

We can express the output function $g$ in Eq. (2) as being composed of two parts. The first part $\varphi_\tau$ is a function that returns the parameter set $\phi_t$ given the hidden state $\mathbf{h}_{t-1}$, i.e., $\phi_t = \varphi_\tau(\mathbf{h}_{t-1})$, while the second part of $g$ returns the density of $\mathbf{x}_t$, i.e., $p_{\phi_t}(\mathbf{x}_t \mid \mathbf{x}_{<t})$.

When modelling high-dimensional and real-valued sequences, a reasonable choice of an observation model is a Gaussian mixture model (GMM) as used in [7]. For GMM, $\varphi_\tau$ returns a set of mixture coefficients $\alpha_t$, means $\boldsymbol{\mu}_{\cdot,t}$ and covariances $\Sigma_{\cdot,t}$ of the corresponding mixture components. The probability of $\mathbf{x}_t$ under the mixture distribution is:

$$p_{\boldsymbol{\alpha}_t, \boldsymbol{\mu}_{\cdot,t}, \Sigma_{\cdot,t}}(\mathbf{x}_t \mid \mathbf{x}_{<t}) = \sum_j \alpha_{j,t} \mathcal{N}\left(\mathbf{x}_t; \boldsymbol{\mu}_{j,t}, \Sigma_{j,t}\right).$$

With the notable exception of [7], there has been little work investigating the structured output density model for RNNs with real-valued sequences.

There is potentially a significant issue in the way the RNN models output variability. Given a deterministic transition function, the only source of variability is in the conditional output probability density. This can present problems when modelling sequences that are at once highly variable and highly structured (i.e., with a high signal-to-noise ratio). To effectively model these types of sequences, the RNN must be capable of mapping very small variations in $\mathbf{x}_t$ (i.e., the only source of randomness) to potentially very large variations in the hidden state $\mathbf{h}_t$. Limiting the capacity of the network, as must be done to guard against overfitting, will force a compromise between the generation of a clean signal and encoding sufficient input variability to capture the high-level variability both within a single observed sequence and across data examples.

The need for highly structured output functions in an RNN has been previously noted. Boulanger-lewandowski et al. [4] extensively tested NADE and RBM-based output densities for modelling sequences of binary vector representations of music. Bayer and Osendorfer [2] introduced a sequence of independent latent variables corresponding to the states of the RNN. Their model, called *STORN*, first generates a sequence of samples $\mathbf{z} = (\mathbf{z}_1, \ldots, \mathbf{z}_T)$ from the sequence of independent latent random variables. At each timestep, the transition function $f$ from Eq. (1) computes the next hidden state $\mathbf{h}_t$ based on the previous state $\mathbf{h}_{t-1}$, the previous output $\mathbf{x}_{t-1}$ and the sampled latent random variables $\mathbf{z}_t$. They proposed to train this model based on the VAE principle (see Sec. 2.2). Similarly, Pachitariu and Sahani [16] earlier proposed both a sequence of independent latent random variables and a stochastic hidden state for the RNN.

These approaches are closely related to the approach proposed in this paper. However, there is a major difference in how the prior distribution over the latent random variable is modelled. Unlike the aforementioned approaches, our approach makes the prior distribution of the latent random variable at timestep $t$ dependent on all the preceding inputs via the RNN hidden state $\mathbf{h}_{t-1}$ (see Eq. (5)). The introduction of temporal structure into the prior distribution is expected to improve the representational power of the model, which we empirically observe in the experiments (See Table 1). However, it is important to note that any approach based on having stochastic latent state is orthogonal to having a structured output function, and that these two can be used together to form a single model.

## 2.2 Variational Autoencoder

For non-sequential data, VAEs [11, 17] have recently been shown to be an effective modelling paradigm to recover complex multimodal distributions over the data space. A VAE introduces a set of latent random variables $\mathbf{z}$, designed to capture the variations in the observed variables $\mathbf{x}$. As an example of a directed graphical model, the joint distribution is defined as:

$$p(\mathbf{x}, \mathbf{z}) = p(\mathbf{x} \mid \mathbf{z})p(\mathbf{z}). \tag{3}$$

The prior over the latent random variables, $p(\mathbf{z})$, is generally chosen to be a simple Gaussian distribution and the conditional $p(\mathbf{x} \mid \mathbf{z})$ is an arbitrary observation model whose parameters are computed by a parametric function of $\mathbf{z}$. Importantly, the VAE typically parameterizes $p(\mathbf{x} \mid \mathbf{z})$ with a highly flexible function approximator such as a neural network. While latent random variable models of the form given in Eq. (3) are not uncommon, endowing the conditional $p(\mathbf{x} \mid \mathbf{z})$ as a potentially highly non-linear mapping from $\mathbf{z}$ to $\mathbf{x}$ is a rather unique feature of the VAE.

However, introducing a highly non-linear mapping from $\mathbf{z}$ to $\mathbf{x}$ results in intractable inference of the posterior $p(\mathbf{z} \mid \mathbf{x})$. Instead, the VAE uses a variational approximation $q(\mathbf{z} \mid \mathbf{x})$ of the posterior that

enables the use of the lower bound:

$$\log p(\mathbf{x}) \geq -\mathrm{KL}(q(\mathbf{z} \mid \mathbf{x}) \| p(\mathbf{z})) + \mathbb{E}_{q(\mathbf{z}|\mathbf{x})} \left[ \log p(\mathbf{x} \mid \mathbf{z}) \right], \tag{4}$$

where $\mathrm{KL}(Q \| P)$ is Kullback-Leibler divergence between two distributions $Q$ and $P$.

In [11], the approximate posterior $q(\mathbf{z} \mid \mathbf{x})$ is a Gaussian $\mathcal{N}(\boldsymbol{\mu}, \mathrm{diag}(\boldsymbol{\sigma}^2))$ whose mean $\boldsymbol{\mu}$ and variance $\boldsymbol{\sigma}^2$ are the output of a highly non-linear function of $\mathbf{x}$, once again typically a neural network.

The generative model $p(\mathbf{x} \mid \mathbf{z})$ and inference model $q(\mathbf{z} \mid \mathbf{x})$ are then trained jointly by maximizing the variational lower bound with respect to their parameters, where the integral with respect to $q(\mathbf{z} \mid \mathbf{x})$ is approximated stochastically. The gradient of this estimate can have a low variance estimate, by reparameterizing $\mathbf{z} = \boldsymbol{\mu} + \boldsymbol{\sigma} \odot \boldsymbol{\epsilon}$ and rewriting:

$$\mathbb{E}_{q(\mathbf{z}|\mathbf{x})} \left[ \log p(\mathbf{x} \mid \mathbf{z}) \right] = \mathbb{E}_{p(\boldsymbol{\epsilon})} \left[ \log p(\mathbf{x} \mid \mathbf{z} = \boldsymbol{\mu} + \boldsymbol{\sigma} \odot \boldsymbol{\epsilon}) \right],$$

where $\boldsymbol{\epsilon}$ is a vector of standard Gaussian variables. The inference model can then be trained through standard backpropagation technique for stochastic gradient descent.

## 3   Variational Recurrent Neural Network

In this section, we introduce a recurrent version of the VAE for the purpose of modelling sequences. Drawing inspiration from simpler dynamic Bayesian networks (DBNs) such as HMMs and Kalman filters, the proposed *variational recurrent neural network* (VRNN) explicitly models the dependencies between latent random variables across subsequent timesteps. However, unlike these simpler DBN models, the VRNN retains the flexibility to model highly non-linear dynamics.

**Generation**   The VRNN contains a VAE at every timestep. However, these VAEs are conditioned on the state variable $\mathbf{h}_{t-1}$ of an RNN. This addition will help the VAE to take into account the temporal structure of the sequential data. Unlike a standard VAE, the prior on the latent random variable is no longer a standard Gaussian distribution, but follows the distribution:

$$\mathbf{z}_t \sim \mathcal{N}(\boldsymbol{\mu}_{0,t}, \mathrm{diag}(\boldsymbol{\sigma}_{0,t}^2)) \text{, where } [\boldsymbol{\mu}_{0,t}, \boldsymbol{\sigma}_{0,t}] = \varphi_\tau^{\mathrm{prior}}(\mathbf{h}_{t-1}), \tag{5}$$

where $\boldsymbol{\mu}_{0,t}$ and $\boldsymbol{\sigma}_{0,t}$ denote the parameters of the conditional prior distribution. Moreover, the generating distribution will not only be conditioned on $\mathbf{z}_t$ but also on $\mathbf{h}_{t-1}$ such that:

$$\mathbf{x}_t \mid \mathbf{z}_t \sim \mathcal{N}(\boldsymbol{\mu}_{x,t}, \mathrm{diag}(\boldsymbol{\sigma}_{x,t}^2)) \text{, where } [\boldsymbol{\mu}_{x,t}, \boldsymbol{\sigma}_{x,t}] = \varphi_\tau^{\mathrm{dec}}(\varphi_\tau^{\mathbf{z}}(\mathbf{z}_t), \mathbf{h}_{t-1}), \tag{6}$$

where $\boldsymbol{\mu}_{x,t}$ and $\boldsymbol{\sigma}_{x,t}$ denote the parameters of the generating distribution, $\varphi_\tau^{\mathrm{prior}}$ and $\varphi_\tau^{\mathrm{dec}}$ can be any highly flexible function such as neural networks. $\varphi_\tau^{\mathbf{x}}$ and $\varphi_\tau^{\mathbf{z}}$ can also be neural networks, which extract features from $\mathbf{x}_t$ and $\mathbf{z}_t$, respectively. We found that these feature extractors are crucial for learning complex sequences. The RNN updates its hidden state using the recurrence equation:

$$\mathbf{h}_t = f_\theta \left( \varphi_\tau^{\mathbf{x}}(\mathbf{x}_t), \varphi_\tau^{\mathbf{z}}(\mathbf{z}_t), \mathbf{h}_{t-1} \right), \tag{7}$$

where $f$ was originally the transition function from Eq. (1). From Eq. (7), we find that $\mathbf{h}_t$ is a function of $\mathbf{x}_{\leq t}$ and $\mathbf{z}_{\leq t}$. Therefore, Eq. (5) and Eq. (6) define the distributions $p(\mathbf{z}_t \mid \mathbf{x}_{<t}, \mathbf{z}_{<t})$ and $p(\mathbf{x}_t \mid \mathbf{z}_{\leq t}, \mathbf{x}_{<t})$, respectively. The parameterization of the generative model results in and – was motivated by – the factorization:

$$p(\mathbf{x}_{\leq T}, \mathbf{z}_{\leq T}) = \prod_{t=1}^{T} p(\mathbf{x}_t \mid \mathbf{z}_{\leq t}, \mathbf{x}_{<t}) p(\mathbf{z}_t \mid \mathbf{x}_{<t}, \mathbf{z}_{<t}). \tag{8}$$

**Inference**   In a similar fashion, the approximate posterior will not only be a function of $\mathbf{x}_t$ but also of $\mathbf{h}_{t-1}$ following the equation:

$$\mathbf{z}_t \mid \mathbf{x}_t \sim \mathcal{N}(\boldsymbol{\mu}_{z,t}, \mathrm{diag}(\boldsymbol{\sigma}_{z,t}^2)) \text{, where } [\boldsymbol{\mu}_{z,t}, \boldsymbol{\sigma}_{z,t}] = \varphi_\tau^{\mathrm{enc}}(\varphi_\tau^{\mathbf{x}}(\mathbf{x}_t), \mathbf{h}_{t-1}), \tag{9}$$

similarly $\boldsymbol{\mu}_{z,t}$ and $\boldsymbol{\sigma}_{z,t}$ denote the parameters of the approximate posterior. We note that the encoding of the approximate posterior and the decoding for generation are tied through the RNN hidden state $\mathbf{h}_{t-1}$. We also observe that this conditioning on $\mathbf{h}_{t-1}$ results in the factorization:

$$q(\mathbf{z}_{\leq T} \mid \mathbf{x}_{\leq T}) = \prod_{t=1}^{T} q(\mathbf{z}_t \mid \mathbf{x}_{\leq t}, \mathbf{z}_{<t}). \tag{10}$$

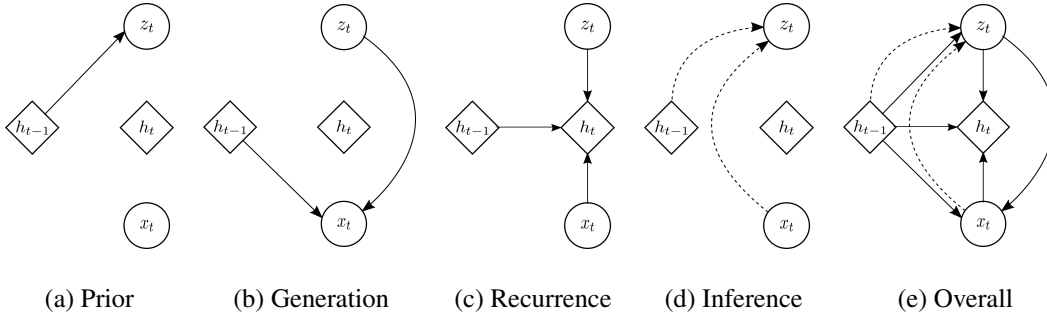

(a) Prior      (b) Generation      (c) Recurrence      (d) Inference      (e) Overall

Figure 1: Graphical illustrations of each operation of the VRNN: (a) computing the conditional prior using Eq. (5); (b) generating function using Eq. (6); (c) updating the RNN hidden state using Eq. (7); (d) inference of the approximate posterior using Eq. (9); (e) overall computational paths of the VRNN.

**Learning**    The objective function becomes a timestep-wise variational lower bound using Eq. (8) and Eq. (10):

$$\mathbb{E}_{q(\mathbf{z}_{\leq T} \mid \mathbf{x}_{\leq T})} \left[ \sum_{t=1}^{T} \left( -\mathrm{KL}(q(\mathbf{z}_t \mid \mathbf{x}_{\leq t}, \mathbf{z}_{<t}) \| p(\mathbf{z}_t \mid \mathbf{x}_{<t}, \mathbf{z}_{<t})) + \log p(\mathbf{x}_t \mid \mathbf{z}_{\leq t}, \mathbf{x}_{<t}) \right) \right]. \qquad (11)$$

As in the standard VAE, we learn the generative and inference models jointly by maximizing the variational lower bound with respect to their parameters. The schematic view of the VRNN is shown in Fig. 1, operations (a)–(d) correspond to Eqs. (5)–(7), (9), respectively. The VRNN applies the operation (a) when computing the conditional prior (see Eq. (5)). If the variant of the VRNN (VRNN-I) does not apply the operation (a), then the prior becomes independent across timesteps. STORN [2] can be considered as an instance of the VRNN-I model family. In fact, STORN puts further restrictions on the dependency structure of the approximate inference model. We include this version of the model (VRNN-I) in our experimental evaluation in order to directly study the impact of including the temporal dependency structure in the prior (i.e., conditional prior) over the latent random variables.

## 4   Experiment Settings

We evaluate the proposed VRNN model on two tasks: (1) modelling natural speech directly from the raw audio waveforms; (2) modelling handwriting generation.

**Speech modelling**    We train the models to directly model raw audio signals, represented as a sequence of 200-dimensional frames. Each frame corresponds to the real-valued amplitudes of 200 consecutive raw acoustic samples. Note that this is unlike the conventional approach for modelling speech, often used in speech synthesis where models are expressed over representations such as spectral features [see, e.g., 18, 3, 13].

We evaluate the models on the following four speech datasets:

1. **Blizzard**: This text-to-speech dataset made available by the Blizzard Challenge 2013 contains 300 hours of English, spoken by a single female speaker [10].
2. **TIMIT**: This widely used dataset for benchmarking speech recognition systems contains $6,300$ English sentences, read by 630 speakers.
3. **Onomatopoeia**[2]: This is a set of $6,738$ non-linguistic human-made sounds such as coughing, screaming, laughing and shouting, recorded from 51 voice actors.
4. **Accent**: This dataset contains English paragraphs read by $2,046$ different native and non-native English speakers [19].

Table 1: Average log-likelihood on the test (or validation) set of each task.

| Models | Speech modelling | | | | Handwriting |
| --- | --- | --- | --- | --- | --- |
| | Blizzard | TIMIT | Onomatopoeia | Accent | IAM-OnDB |
| RNN-Gauss | 3539 | -1900 | -984 | -1293 | 1016 |
| RNN-GMM | 7413 | 26643 | 18865 | 3453 | 1358 |
| VRNN-I-Gauss | $\geq$ 8933 | $\geq$ 28340 | $\geq$ 19053 | $\geq$ 3843 | $\geq$ 1332 |
| | $\approx$ 9188 | $\approx$ 29639 | $\approx$ 19638 | $\approx$ 4180 | $\approx$ 1353 |
| VRNN-Gauss | $\geq$ 9223 | $\geq$ 28805 | $\geq$ 20721 | $\geq$ 3952 | $\geq$ 1337 |
| | $\approx$ **9516** | $\approx$ **30235** | $\approx$ **21332** | $\approx$ 4223 | $\approx$ 1354 |
| VRNN-GMM | $\geq$ 9107 | $\geq$ 28982 | $\geq$ 20849 | $\geq$ 4140 | $\geq$ 1384 |
| | $\approx$ 9392 | $\approx$ 29604 | $\approx$ 21219 | $\approx$ **4319** | $\approx$ **1384** |

For the Blizzard and Accent datasets, we process the data so that each sample duration is $0.5s$ (the sampling frequency used is 16kHz). Except the TIMIT dataset, the rest of the datasets do not have predefined train/test splits. We shuffle and divide the data into train/validation/test splits using a ratio of $0.9/0.05/0.05$.

**Handwriting generation**   We let each model learn a sequence of $(x, y)$ coordinates together with binary indicators of pen-up/pen-down, using the IAM-OnDB dataset, which consists of $13,040$ handwritten lines written by $500$ writers [14]. We preprocess and split the dataset as done in [7].

**Preprocessing and training**   The only preprocessing used in our experiments is normalizing each sequence using the global mean and standard deviation computed from the entire training set. We train each model with stochastic gradient descent on the negative log-likelihood using the Adam optimizer [12], with a learning rate of $0.001$ for TIMIT and Accent and $0.0003$ for the rest. We use a minibatch size of $128$ for Blizzard and Accent and $64$ for the rest. The final model was chosen with early-stopping based on the validation performance.

**Models**   We compare the VRNN models with the standard RNN models using two different output functions: a simple Gaussian distribution (Gauss) and a Gaussian mixture model (GMM). For each dataset, we conduct an additional set of experiments for a VRNN model without the conditional prior (VRNN-I).

We fix each model to have a single recurrent hidden layer with $2000$ LSTM units (in the case of Blizzard, $4000$ and for IAM-OnDB, $1200$). All of $\varphi_\tau$ shown in Eqs. (5)–(7), (9) have four hidden layers using rectified linear units [15] (for IAM-OnDB, we use a single hidden layer). The standard RNN models only have $\varphi_\tau^{\mathbf{x}}$ and $\varphi_\tau^{\mathrm{dec}}$, while the VRNN models also have $\varphi_\tau^{\mathbf{z}}$, $\varphi_\tau^{\mathrm{enc}}$ and $\varphi_\tau^{\mathrm{prior}}$. For the standard RNN models, $\varphi_\tau^{\mathbf{x}}$ is the feature extractor, and $\varphi_\tau^{\mathrm{dec}}$ is the generating function. For the RNN-GMM and VRNN models, we match the total number of parameters of the deep neural networks (DNNs), $\varphi_\tau^{\mathbf{x},\mathbf{z},\mathrm{enc},\mathrm{dec},\mathrm{prior}}$, as close to the RNN-Gauss model having $600$ hidden units for every layer that belongs to either $\varphi_\tau^{\mathbf{x}}$ or $\varphi_\tau^{\mathrm{dec}}$ (we consider $800$ hidden units in the case of Blizzard). Note that we use 20 mixture components for models using a GMM as the output function.

For qualitative analysis of speech generation, we train larger models to generate audio sequences. We stack three recurrent hidden layers, each layer contains $3000$ LSTM units. Again for the RNN-GMM and VRNN models, we match the total number of parameters of the DNNs to be equal to the RNN-Gauss model having $3200$ hidden units for each layer that belongs to either $\varphi_\tau^{\mathbf{x}}$ or $\varphi_\tau^{\mathrm{dec}}$.

## 5   Results and Analysis

We report the average log-likelihood of test examples assigned by each model in Table 1. For RNN-Gauss and RNN-GMM, we report the exact log-likelihood, while in the case of VRNNs, we report the variational lower bound (given with $\geq$ sign, see Eq. (4)) and approximated marginal log-likelihood (given with $\approx$ sign) based on importance sampling using $40$ samples as in [17]. In general, higher numbers are better. Our results show that the VRNN models have higher log-likelihood, which support our claim that latent random variables are helpful when modelling com-

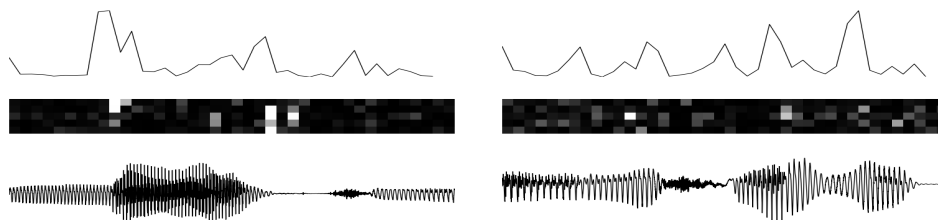

Figure 2: The top row represents the difference $\delta_t$ between $\boldsymbol{\mu}_{z,t}$ and $\boldsymbol{\mu}_{z,t-1}$. The middle row shows the dominant KL divergence values in temporal order. The bottom row shows the input waveforms.

plex sequences. The VRNN models perform well even with a unimodal output function (VRNN-Gauss), which is not the case for the standard RNN models.

**Latent space analysis**   In Fig. 2, we show an analysis of the latent random variables. We let a VRNN model read some unseen examples and observe the transitions in the latent space. We compute $\delta_t = \sum_j (\boldsymbol{\mu}_{z,t}^j - \boldsymbol{\mu}_{z,t-1}^j)^2$ at every timestep and plot the results on the top row of Fig. 2. The middle row shows the KL divergence computed between the approximate posterior and the conditional prior. When there is a transition in the waveform, the KL divergence tends to grow (white is high), and we can clearly observe a peak in $\delta_t$ that can affect the RNN dynamics to change modality.

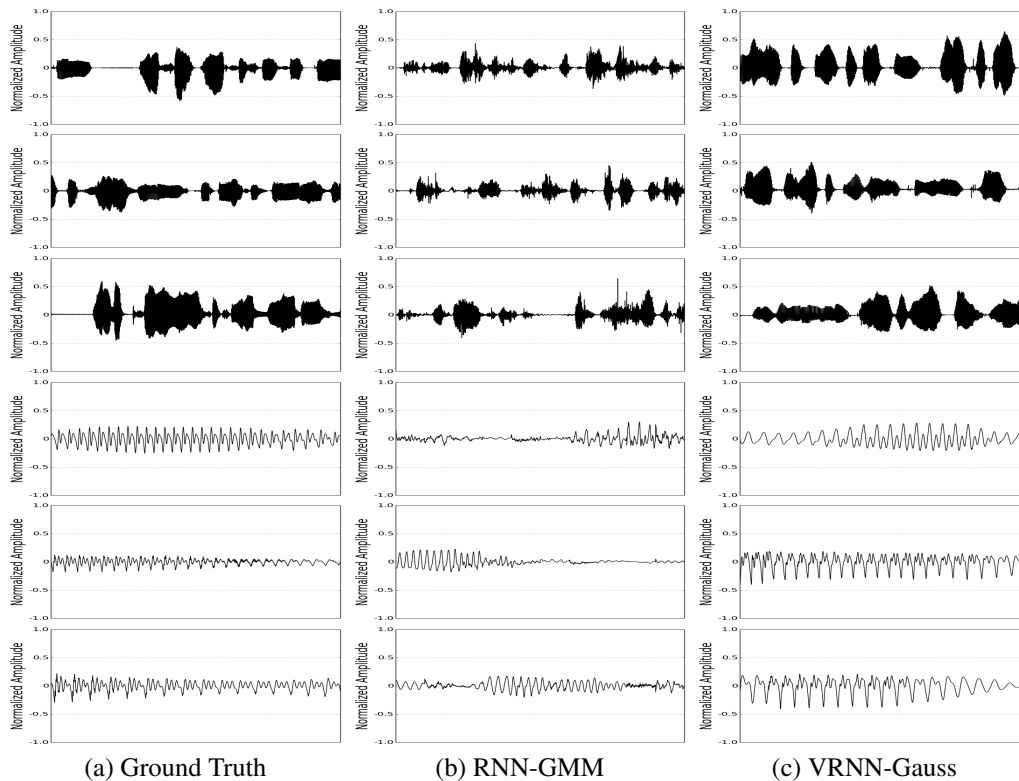

(a) Ground Truth          (b) RNN-GMM          (c) VRNN-Gauss

Figure 3: Examples from the training set and generated samples from RNN-GMM and VRNN-Gauss. Top three rows show the global waveforms while the bottom three rows show more zoomed-in waveforms. Samples from (b) RNN-GMM contain high-frequency noise, and samples from (c) VRNN-Gauss have less noise. We exclude RNN-Gauss, because the samples are almost close to pure noise.

**Speech generation**   We generate waveforms with $2.0s$ duration from the models that were trained on Blizzard. From Fig. 3, we can clearly see that the waveforms from the VRNN-Gauss are much less noisy and have less spurious peaks than those from the RNN-GMM. We suggest that the large amount of noise apparent in the waveforms from the RNN-GMM model is a consequence of the compromise these models must make between representing a clean signal consistent with the training data and encoding sufficient input variability to capture the variations across data examples. The latent random variable models can avoid this compromise by adding variability in the latent space, which can always be mapped to a point close to a relatively clean sample.

**Handwriting generation**   Visual inspection of the generated handwriting (as shown in Fig. 4) from the trained models reveals that the VRNN model is able to generate more diverse writing style while maintaining consistency within samples.

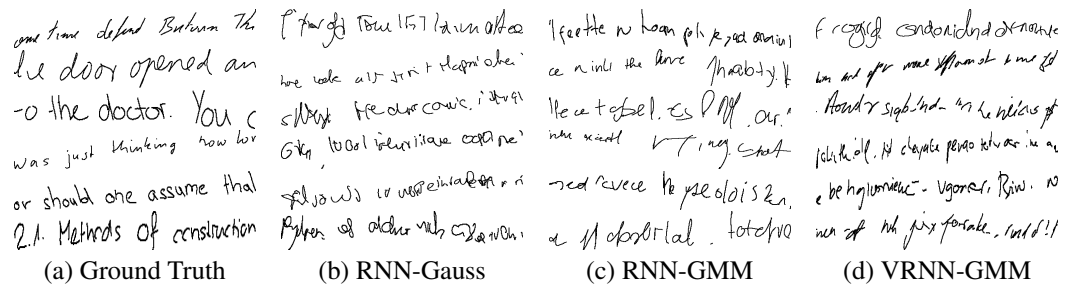

|                   |                 |               |                  |
|-------------------|-----------------|---------------|------------------|
| (a) Ground Truth  | (b) RNN-Gauss   | (c) RNN-GMM   | (d) VRNN-GMM     |

Figure 4: Handwriting samples: (a) training examples and unconditionally generated handwriting from (b) RNN-Gauss, (c) RNN-GMM and (d) VRNN-GMM. The VRNN-GMM retains the writing style from beginning to end while RNN-Gauss and RNN-GMM tend to change the writing style during the generation process. This is possibly because the sequential latent random variables can guide the model to generate each sample with a consistent writing style.

## 6   Conclusion

We propose a novel model that can address sequence modelling problems by incorporating latent random variables into a recurrent neural network (RNN). Our experiments focus on unconditional natural speech generation as well as handwriting generation. We show that the introduction of latent random variables can provide significant improvements in modelling highly structured sequences such as natural speech sequences. We empirically show that the inclusion of randomness into high-level latent space can enable the VRNN to model natural speech sequences with a simple Gaussian distribution as the output function. However, the standard RNN model using the same output function fails to generate reasonable samples. An RNN-based model using more powerful output function such as a GMM can generate much better samples, but they contain a large amount of high-frequency noise compared to the samples generated by the VRNN-based models.

We also show the importance of temporal conditioning of the latent random variables by reporting higher log-likelihood numbers on modelling natural speech sequences. In handwriting generation, the VRNN model is able to model the diversity across examples while maintaining consistent writing style over the course of generation.

## Acknowledgments

The authors would like to thank the developers of Theano [1]. Also, the authors thank Kyunghyun Cho, Kelvin Xu and Sungjin Ahn for insightful comments and discussion. We acknowledge the support of the following agencies for research funding and computing support: Ubisoft, NSERC, Calcul Québec, Compute Canada, the Canada Research Chairs and CIFAR.

## Footnotes

[1]Code is available at `http://www.github.com/jych/nips2015_vrnn`

[2] This dataset has been provided by Ubisoft.

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
