[Supplementary Material · vrnn_appendix.pdf]

# Supplementary Material: A Recurrent Latent Variable Model for Sequential Data

**Junyoung Chung,  Kyle Kastner,  Laurent Dinh,  Kratarth Goel,**
**Aaron Courville,  Yoshua Bengio**[*]
Department of Computer Science and Operations Research
Université de Montréal
[*]CIFAR Senior Fellow
{firstname.lastname}@umontreal.ca

## A    Derivation of the Variational Lower Bound

The structure of the approximate posterior follows the structure of the prior and allows us to have the following decomposition

$$\int q(\mathbf{z}_{\leq T} \mid \mathbf{x}_{\leq T}) \log \left( \frac{p(\mathbf{x}_{\leq T}, \mathbf{z}_{\leq T})}{q(\mathbf{z}_{\leq T} \mid \mathbf{x}_{\leq T})} \right) d\mathbf{z}_{\leq T}$$

$$= \int \sum_{t=1}^{T} \left( q(\mathbf{z}_{\leq T} \mid \mathbf{x}_{\leq T}) \log \left( \frac{p(\mathbf{z}_t \mid \mathbf{x}_{<t}, \mathbf{z}_{<t}) p(\mathbf{x}_t \mid \mathbf{z}_{\leq t}, \mathbf{x}_{<t})}{q(\mathbf{z}_t \mid \mathbf{x}_{\leq t}, \mathbf{z}_{<t})} \right) \right) d\mathbf{z}_{\leq T}$$

$$= \sum_{t=1}^{T} \left( \int q(\mathbf{z}_{\leq t} \mid \mathbf{x}_{\leq t}) \log \left( \frac{p(\mathbf{z}_t \mid \mathbf{x}_{<t}, \mathbf{z}_{<t}) p(\mathbf{x}_t \mid \mathbf{z}_{\leq t}, \mathbf{x}_{<t})}{q(\mathbf{z}_t \mid \mathbf{x}_{\leq t}, \mathbf{z}_{<t})} \right) d\mathbf{z}_{\leq t} \right)$$

$$= \sum_{t=1}^{T} \left( \int q(\mathbf{z}_{\leq t} \mid \mathbf{x}_{\leq t}) \log p(\mathbf{x}_t \mid \mathbf{z}_{\leq t}, \mathbf{x}_{<t}) \, d\mathbf{z}_{\leq t} + \int q(\mathbf{z}_{\leq t} \mid \mathbf{x}_{\leq t}) \log \left( \frac{p(\mathbf{z}_t \mid \mathbf{x}_{<t}, \mathbf{z}_{<t})}{q(\mathbf{z}_t \mid \mathbf{x}_{\leq t}, \mathbf{z}_{<t})} \right) d\mathbf{z}_{\leq t} \right)$$

$$= \sum_{t=1}^{T} \left( \int q(\mathbf{z}_{\leq t} \mid \mathbf{x}_{\leq t}) \log p(\mathbf{x}_t \mid \mathbf{z}_{\leq t}, \mathbf{x}_{<t}) \, d\mathbf{z}_{\leq t} - \int q(\mathbf{z}_{<t} \mid \mathbf{x}_{<t}) \operatorname{KL}(q(\mathbf{z}_t \mid \mathbf{x}_{\leq t}, \mathbf{z}_{<t}) \| p(\mathbf{z}_t \mid \mathbf{x}_{<t}, \mathbf{z}_{<t})) \, d\mathbf{z}_{<t} \right)$$

$$= \mathbb{E}_{\mathbf{z}_{\leq T} \sim q(\mathbf{z}_{\leq T} \mid \mathbf{x}_{\leq T})} \left[ \sum_{t=1}^{T} \left( -\operatorname{KL}(q(\mathbf{z}_t \mid \mathbf{x}_{\leq t}, \mathbf{z}_{<t}) \| p(\mathbf{z}_t \mid \mathbf{x}_{<t}, \mathbf{z}_{<t})) + \log p(\mathbf{x}_t \mid \mathbf{z}_{\leq t}, \mathbf{x}_{<t}) \right) \right]$$

$$\simeq \sum_{t=1}^{T} \left( \log p(\mathbf{x}_t \mid \mathbf{z}_{\leq t}, \mathbf{x}_{<t}) - \operatorname{KL}(q(\mathbf{z}_t \mid \mathbf{x}_{\leq t}, \mathbf{z}_{<t}) \| p(\mathbf{z}_t \mid \mathbf{x}_{<t}, \mathbf{z}_{<t})) \right) \text{ where } \mathbf{z}_{\leq T} \sim q(\mathbf{z}_{\leq T} \mid \mathbf{x}_{\leq T})$$