[Reviews · NeurIPS 2015]

Submitted by Assigned_Reviewer_1

This paper investigates the application of the variational autoencoder to a recurrent model. The work is well motivated in the introduction, then the paper becomes less and less clear (for instance equations 1 and 2 have a problem). The real contribution starts in the middle of page 4.

Note that the first two sentences of the paragraph "Generation" in this page is the true summary of the paper. These two sentences could appear earlier in the paper, even in a different version.

However, the main issue comes with the experimental part and table 1. The evaluation criterion is the average log-probability. If this really the case, all results in table 1 should be negative, not only the 3 first scores of the first row. Otherwise, maybe it's the minus Log probability, but in this case the lower is the better. At the end, I cannot understand the numbers in this table. Note that these are the only results.

Comments (reading order):

It is true that summarizing the paper of Kingma and Welling, 2014 in less than one page is difficult. However, the result is not always easy to read. It could be maybe easier to directly describe the VAE in the recurrent context.

Page 2: "extend the variational autoencoder (VAE) into *an* recurrent"

" offers a interesting combination highly " -> " offers *an* interesting combination *of* highly ..." Note that this sentence is very long with too much "and".

There is maybe an issue with with these two equations: xt depends on ht in (1) and then in (2) ht depends on xt.

There must be a "t-1" somewhere to avoid this cycle. Is that correct ?

Page 6: "We use truncated backpropagation through time and initialize hidden state with the final hidden state of previous mini batch,

resetting to a zero-vector every four updates." This part sounds tricky. Could you comment if this necessary because for the conventional model or for the one with latente variables, or both ?
Summary: The work is well motivated in the introduction. The idea to apply the variational autoencoder at each time step of a recurrent network is really nice. However, there is a problem with the experimental results that I cannot understand.

Submitted by Assigned_Reviewer_2

This paper empirically shows how adding latent variables to an RNN improves its generative power by showing nice samples and better likelihoods (or lower bound on likelihoods) than previous attempts at generating distributions over continuous variables.

The paper is well written and clear - except for the figure, which is quite confusing and confused me for a while. As the authors admit, their model is a slight modification to e.g. [1]. The authors did, as far as one can tell, a fair comparison with the model presented in [1], and showed how adding more structure to the prior over latent variables z_t (by means of making the mean / variances of those a function of the previous hidden state) helped generation.

My main concern about the paper is that is rather incremental. Given that the main novelty involves a better prior over z_t, I would have liked to see more analysis on why that helped (other than through empirical evaluation). Also, unlike previous work, the authors seem to use the same hidden state h_t at generation and for inference. What is the motivation behind this?

Regarding the experiments with speech, could the authors clarify how exactly the windowing was made for the 200 waveform samples (was it overlapping from sample to sample?).

The authors could reference DRAW as another application / use case of a VRNN-like architecture.
Summary: Despite the fact that adding latent variables to RNNs has already been explored in previous (recent) work, this paper is well written and presents a slight modification that outperforms all previous attempts at integrating the variational AE framework with RNNs.

Submitted by Assigned_Reviewer_3

The paper introduces a generative model for sequences. The novelty consists in considering explicit latent stochastic variables that are used to condition the data production. These latent variables may depend on the past.

Pro: provides an extension of recent latent variable models to sequences.

They show that the model is competitive with other sequence models on a series of datasets.

Cons: some claims are not fully supported. The experiments are only proof of concepts. They just used raw signal and no pre-preprocessing

at all, which makes it difficult to compare with state of the art performance on these classical datasets.
Summary: aa

Submitted by Assigned_Reviewer_4

This paper presents a variational inference approach to learn Recurrent Neural Networks (RNNs) that have stochastic transitions between hidden states. The model is a stochastic nonlinear dynamical system similar to a nonlinear state-space model where the noise between hidden states (i.e. the process noise) is not identically distributed but depends on the hidden state. The parameters of the generative model are learnt simultaneously with a recognition model via maximization of the evidence lower bound.

This is a solid paper which bridges the gap between RNNs and nonlinear state-space models. The presentation is clear with an emphasis on presenting the general ideas rather than the technical details. The model itself is simple to describe and the variational training procedure is now standard. There is probably an important engineering effort to achieve the reported results which is not obvious when reading the paper. Also there is no mention of computing machinery or training times.

The notation with subindices "< t" sometimes hides the Markovian structure of the model which can be useful when interpreting the factorizations of joint distributions.

The literature review does not seem to mention nonlinear state-space models (i.e. nonlinear versions of linear state-space models aka "Kalman filters"). Those models are popular in engineering, robotics, neuroscience and many other fields. The model presented in this paper could have an impact on fields beyond machine learning. It could be useful to be more explicit in the connection with nonlinear state-space models.

A few questions:

If VRNN-Gauss was not able to generate well-formed handwritting, are all the plots labelled VRNN actually VRNN-GMM?

Is there any regularization applied to the model? If not, how was overfitting prevented?

Equation (1) gives h_t as a function of x_t while Equation (2) gives a distribution over x_t as a function of h_t. Is this intended?
Summary: This is a solid paper which bridges the gap between RNNs and nonlinear state-space models. The experiments are convincing and the presentation is clear and well polished.

Author Feedback
Author rebuttal: Assigned_reviewer_1 :
We will redraw the figures and maintain wav files of generated samples on a project webpage or github.

Assigned_reviewer_2 :
"My ... incremental." - Previous studies combining an RNN and VAE [1,2] are only using toy datasets, and there is no evidence that these models will scale to larger datasets such as 300 hour wav files (Blizzard). Also the idea of incorporating variational learning into the RNN is similar, but the interpretation of latent variables is quite different. Structuring the latent variables in the way we prescribe is helpful to learn sequential data which contains temporal structure. There are techniques which are not shown in previous studies such as sharing an RNN for inference and generation, which can work equally well or even better than separate RNN encoder/decoder pairs, while also reducing required memory. We found having a deep neural net feature extractors are crucial to model raw speech signals. These feature extractors were shared across different procedures: inference, generation, updating recurrence.

"Also, unlike ... behind this" - The purpose of sharing an RNN, is to regularize the parameters of the RNN in order to generalize better. By sharing the RNN, we can reduce the number of parameters, reducing the memory required for the model and also obtaining better performance than not sharing.

"Regarding ... samples." - There were no overlaps between frames. In the currently described procedure, having overlaps between frames can yield bad results because phase is difficult to predict exactly, and small mismatches in phase will deteriorate the predicted samples when using overlap-add.

"The ... DRAW ... architect" - Our main concern was applying RNN with latent variables on 'sequential' data, so DRAW was excluded since it worked on non-sequential data (though performing a sequential inference and generation procedure). We will cite DRAW.

Assigned_reviewer_3 :
"If VRNN- ... VRNN-GMM" - They are samples from VRNN-GMM, and we will make this more clear in the surrounding narrative.

"Is ... prevented" - For the Blizzard dataset, there are more than two millions of training examples (of 0.5 second each) so in that case we didn't need to fight with overfitting. Otherwise we use the typical procedures such as reducing the model capacity, using weight norm regularization, adding weight noise and early stopping on validation error.

"Eq. (1) ... intended" - It was a typo, and should be h_{t-1} instead of h_t.

Assigned_reviewer_4 :
"There is ... correct?" - Eq. (2) had a typo, it should be h_t -> h_{t-1}.

"If this really ... in this table." - The term log-probability triggered confusion, we will correct it as average log-likelihood (LL). The log-likelihood for continuous data is the sum of the log densities, evaluated at every observation. While it's true for discrete set that probability mass function has to be in the [0, 1] range, it is not *uncommon* for density function to reach points higher than 1 (and have positive log-density at these points). In fact, almost any distribution with a small enough scale can have a density function that reaches a maximum higher than 1. For example, any Gaussian with a precision higher than 2 {\pi} will have a positive log-density at its mode. Therefore, log-likelihood of a sequence could become a positive number. We report the log-likelihood instead of the negative case. You can also find in [3] (NIPS, 2014) and [4,5] (NIPS, 2013) reporting positive numbers, and 'higher is better'. We apologize that log-probability seems unclear, and as this can confuse readers, we will fix it to log-density or LL.

We will fix the typos as reviewer_4 pointed out.

Truncated backprop through time (TBPTT) was used for Blizzard & Accent datasets, since the raw data contains very long sound clips. It is a technique to learn sequences by breaking a sequence into subsequences and updating more frequently, but make sure they are linked by passing the context. It was dependent on which dataset was being used for training and applied to both RNN/VRNN. We will make this part more clear.

Assigned_reviewer_5 :
"While ... to" - To the best of our knowledge, RNN-RBM has only been successful in the context of binary data. We tried training RNN-GRBM on these continuous tasks as another model comparison, but did not achieve useful performance on any task. Instead, we will include experiment of VRNN tested on discrete tasks.

Assigned_reviewer_6 :
"They ... classical datasets" - To the best of our knowledge, we are the first latent variable RNN publication to generate "real-world" continuous sound sequences instead of toy data.

[1] J. Bayer and C. Osendofer, Learning Stochastic Recurrent Networks
[2] O. Fabius and J. Amersfoort, Variational Recurrent Auto-Encoders
[3] I. Goodfellow et al., Generative Adversarial Nets
[4] B. Uria et al., RNADE: ...
[5] Y. Tang and R. Salakhutdinov, Learning Stochastic Feedforward Neural Networks